# An In Vitro Model to Investigate the Role of *Helicobacter Pylori* in Type 2 Diabetes, Obesity, Alzheimer’s Disease and Cardiometabolic Disease

**DOI:** 10.3390/ijms21218369

**Published:** 2020-11-08

**Authors:** Paola Cuomo, Marina Papaianni, Clementina Sansone, Antonio Iannelli, Domenico Iannelli, Chiara Medaglia, Debora Paris, Andrea Motta, Rosanna Capparelli

**Affiliations:** 1Department of Agriculture Sciences, University of Naples “Federico II”, via Università, 100-Portici, 80055 Naples, Italy; paola.cuomo@unina.it (P.C.); marina.papaianni@unina.it (M.P.); 2Department of Marine Biotechnology, Stazione Zoologica Anton Dohrn, 80121 Naples, Italy; clementina.sansone@szn.it; 3Department of Digestive Surgery, Université Côte d’Azur, Campus Valrose, Batiment L, Avenue de Valrose, 28-CEDEX 2, 06108 Nice, France; iannelli.a@chu-nice.fr; 4Inserm, U1065, Team 8 “Hepatic Complications of Obesity and Alcohol”, Route Saint Antoine de Ginestière 151, BP 2 3194, CEDEX 3, 06204 Nice, France; 5Department of Microbiology and Molecular Medicine, University of Geneva Medical School, rue du Général-Dufour, 1211 Genève, Switzerland; chiara.medaglia@unige.ch; 6Institute of Biomolecular Chemistry, National Research Council, via Campi Flegrei, 34-Pozzuoli, 80078 Naples, Italy; dparis@icb.cnr.it (D.P.); andrea.motta@icb.cnr.it (A.M.)

**Keywords:** *Helicobacter pylori*, mTORC1, branched chain amino acids, inflammation, mitochondrial dysfunction

## Abstract

*Helicobacter pylori* (*Hp*) is a Gram-negative bacterium colonizing the human stomach. Nuclear Magnetic Resonance (NMR) analysis of intracellular human gastric carcinoma cells (MKN-28) incubated with the *Hp* cell filtrate (*Hpcf*) displays high levels of amino acids, including the branched chain amino acids (BCAA) isoleucine, leucine, and valine. Polymerase chain reaction (PCR) Array Technology shows upregulation of mammalian Target Of Rapamycin Complex 1 (mTORC1), inflammation, and mitochondrial dysfunction. The review of literature indicates that these traits are common to type 2 diabetes, obesity, Alzheimer’s diseases, and cardiometabolic disease. Here, we demonstrate how *Hp* may modulate these traits. *Hp* induces high levels of amino acids, which, in turn, activate mTORC1, which is the complex regulating the metabolism of the host. A high level of BCAA and upregulation of mTORC1 are, thus, directly regulated by *Hp*. Furthermore, *Hp* modulates inflammation, which is functional to the persistence of chronic infection and the asymptomatic state of the host. Finally, in order to induce autophagy and sustain bacterial colonization of gastric mucosa, the *Hp* toxin VacA localizes within mitochondria, causing fragmentation of these organelles, depletion of ATP, and oxidative stress. In conclusion, our in vitro disease model replicates the main traits common to the above four diseases and shows how *Hp* may potentially manipulate them.

## 1. Introduction

*Helicobacter pylori (Hp)* is a Gram-negative bacterium colonizing the human gastric mucosa. It is transmitted orally, often within the family. Colonizing about one half of the world’s population, *Hp*, can be classified as one of the most successful human pathogens [1,2]. Genetic diversity and efficient evasion of host innate and adaptive immune responses contribute to this success [3]. Genetic diversity of *Hp* originates from a high mutation rate, and is favored by the absence of a DNA mismatch repair mechanism and acquisition of DNA from other strains [3]. The *Hp* lipopolysaccharide escapes binding by antimicrobial peptides and detection by Tool-like receptors (TLRs) by removing phosphate groups from the lipid A [4]. VacA modulates T-cell proliferation by inhibiting nuclear translocation of the T transcription factor NF-AT [5]. *Hp* can persist for decades asymptomatically, but carriers are at risk of developing gastritis, duodenal ulcers, non-Hodgkin’s lymphoma, or gastric adenocarcinoma. Finally, epidemiological studies suggest that *Hp* may cause extra gastric diseases [6].

Often, bacteria colonize the host by exploiting its metabolism [7]. For this purpose, *Hp* uses mTORC1, which is the complex sensing the conditions for cellular survival. mTORC1 is active when nutrients are abundant, and inactive when they are scarce [8]. In the early phase of host colonization, the *Hp* toxin VacA inhibits mTORC1 expression and activates autophagy to gain the nutrients needed to colonize the gastric mucosa [9]. Once this phase is terminated, autophagy would obstruct the chronic infection. To prevent this condition, CagA (a toxin antagonistic to VacA) activates mTORC1 [10]. The branched chain amino acids (BCAA) leucine, isoleucine, and valine, together with insulin, actively participate to mTORC1 activation [11]. *Hp*, through a clever use of two antagonistic toxins, modulates mTORC1 and, consequently, the metabolism of its host. Furthermore, in order to induce autophagy and sustain colonization of gastric mucosa, VacA localizes within the mitochondria by causing mitochondrial DNA mutations in the gastric cells, mitochondrial fragmentation, depletion of ATP, and oxidative stress [12]. Finally, to facilitate chronic infection and persistence of the host asymptomatic state, *Hp* modulates inflammation. However, this unstable equilibrium often fails. Inflammation is the most frequent cause of the passage from the asymptomatic state to that of disease [13].

We just described how upregulation of mTORC1, high levels of BCAA, inflammation, and mitochondrial dysfunctions characterize *Hp* infection. According to current literature, the same traits characterize the metabolic diseases known as type 2 diabetes (T2D), obesity (OB), Alzheimer’s disease (AD), and cardiometabolic disease (CMD) [14,15,16,17]. It seemed, therefore, feasible to investigate whether *Hp* may potentially predispose to the above diseases.

Epidemiological studies have several limitations. They require the enrollment of a large number of patients, and known and unknown confounding factors make results difficult to replicate [18]. Analysis of the cellular metabolic profile of in vitro cultured cells has detected yeast mutants that conventional methods failed to identify [19], while the medium from cultured human muscle cells identified creatine as a biomarker of human mitochondrial disease [20]. Inspired by these results, we tested whether nuclear magnetic resonance (NMR) metabolomics and microarray analysis of the human gastric carcinoma cells MKN-28, incubated for 2 h with the *Hp* cell filtrate (*Hpcf*), might identify the traits that current literature recognizes as biomarkers of patients with one of the four diseases listed above. MKN-28 incubated with *Hpcf* showed high levels of BCAA, upregulation of mTORC1, mitochondrial dysfunction, and inflammation. Here, we show that these traits are all referable, at least in part, to *Hp*.

## 2. Results

### 2.1. NMR-Based Metabolomics Analysis

The in vitro metabolic effects of *Hpcf* on the MKN-28 cells were investigated by performing NMR-based intracellular and extracellular metabolomics analysis of the MKN-28 cells. Intracellular analysis compared MKN-28 cells incubated and not with the *Hpcf* of the culture medium used to grow *Hp*. We first applied the unsupervised Principal Component Analysis (PCA) to verify the homogeneity of samples and then the supervised orthogonal projection to latent structure analysis (OPLS-DA) to explore the intracellular metabolic profiles. The score plot (Figure 1A) shows a distinct separation between the two sample classes along the first component t[1].

In particular, the extracts of MKN-28 cells incubated with *Hpcf* are placed at t[1] positive coordinates, while the extracts of the MKN-28 cells non-incubated with *Hpcf* are placed at t[1] negative coordinates. To discriminate between the two groups of samples, we used the NMR loadings plot variables with correlation loading values |*p*(corr)| > 0.7 (Figure 1B). Cells incubated with *Hpcf* when compared with non-incubated cells show higher levels of the amino acids phenylalanine, tyrosine, glutamate, proline, leucine, alanine, valine, and isoleucine, the short chain fatty acids acetate and propionate, saturated fatty acids, sialic acid, and lower levels of taurine (Figure 2A and Figure 3A,B).

### 2.2. Helicobacter Pylori Alters the Amino Acids Metabolism

The literature describes a study where two independent cohorts with each consisting of more than 3000 normoglycemic participants being followed for 12 years. High plasma levels of BCAA, and of the two aromatic amino acids (AAA) tyrosine and phenylalanine (out of 60 metabolites) could predict the development of T2D in the enrolled participants as early as 12 years in advance from the onset of the disease [21]. Numerous independent studies confirm the association of high plasma levels of BCAA with T2D [22] and OB [23].

Incubated with *Hpcf* for 2 h, the human MKN-28 gastric cells show altered expression levels of 13 metabolites (Figure 2A and Figure 3A,B) including the five amino acids listed above, which are all expressed at high levels (Figure 2A). These findings, obtained using cells cultured in vitro, replicate the results of the studies carried out on humans and cited above. Furthermore, a recent longitudinal study carried out on a large Japanese population demonstrates that, in addition to the above amino acids, alanine and proline, which are positively associated with visceral fat deposition, while glycine is negatively associated [24]. Our study replicates these results (Figure 2A). The probability that seven out of the 20 (0.35) amino acids upregulated in the studies reported above, and, in the present one, are instead upregulated by chance at 8 × 10^−8^.

Mammals lack the enzymes needed for the synthesis of BCAA [14]. The increased plasma levels of BCAA detected in patients with T2D or OB may result from two potential mechanisms. One claims that an excess of dietary BCAA activates the mTORC1 complex. The alternative mechanism suggests that high levels of BCAA alter the metabolism by causing mitochondrial dysfunction of pancreatic islet β cells [22]. MKN-28 cells incubated with *Hpcf* display upregulation of mTORC1 and mitochondrial dysfunction (Table 1). Therefore, these data indicate that cell culture may fruitfully be used to investigate whether *Hp* influences apparently different diseases. Zucker-obese rats display lower muscle glycine levels compared to Zucker-lean rats. However, when Zucker-obese rats are fed on a BCAA-restricted diet, their muscle glycine levels become normal. These results argue that BCAA metabolism may somehow interfere with that of glycine and, at the same time, give plausibility to the negative association of glycine with BCAA reported above.

BCAA and glutamate are the two pathways more often altered in patients with AD [25]. High levels of isoleucine are associated with AD and high levels of valine are associated with reduced risk of AD. The latter result has been confirmed in the longitudinal Rotterdam study [26]. BCAA cross the blood-brain barrier through the large neutral amino acid transporter LAT1 in competition with AAA [27]. When the plasma level of BCAA is chronically elevated, BCAA uptake in the brain is favored at the expense of tryptophan, which is the precursor of serotonin and a molecule with multiple functions. It reduces amyloid-β (Aβ) production (the hallmark of AD), protects neural survival, and stabilizes the mood [28]. Furthermore, the enzyme-branched chain amino acid transaminase (BCAT) converts BCAA to glutamate. Heightened levels of glutamate cause neuronal death by excitotoxicity (nerve cells death by glutamate overactivation). Finally, mTORC1 activation and mitochondrial dysfunction have a role in AD. The former enhances the *tau*-induced neurodegeneration, and apoptosis of post-mitotic neurons while the latter provokes Aβ plaque formation [29]. In conclusion, high levels of isoleucine, valine, and glutamate (Figure 2A), mTORC1 activation (Table 1), and mitochondrial dysfunction (Table 1) reported in this study successfully recapitulate in vitro what is described in human studies. Of note, the convergent dysmetabolism (high levels of isoleucine, valine, and glutamate) common to T2D, OB, and AD is confirmed by the efficacy in AD patients of drugs used to treat T2D and other metabolic diseases [14].

In addition to T2D, OB, and AD, recently high levels of BCAA have also been shown to have a critical role in the pathogenesis of heart failure [15]. Studies in a mouse model have demonstrated that accumulation of BCAA requires inhibition of the BCAA-degradation-gene Kruppel-like factor 15 (KLF15). The subsequent intra-myocardial accumulation of BCAA upregulates mTORC1 that activates protein synthesis, cardiac hypertrophy, and heart failure. In a mouse model, pharmacologically induced BCAA catabolism significantly re-establishes the cardiac function [30]. Significantly, suppression of BCAA catabolism and accumulation of BCAA have also been observed in humans with heart failure. With regard to mitochondrial function, accumulation of lipids in the heart and hyperglycemia lead to impaired mitochondrial phosphorylation [31].

### 2.3. MKN-28 Cells Uptake BCAA from Culture Medium

The four diseases listed above have high levels of BCAA as a common feature. Mammalian cells as well as *Hp* cannot synthesize essential amino acids (which include BCAA). To shed light on the origin of the high levels of BCAAs observed in our experiments, we tested the hypothesis that they may derive from the depletion of culture medium by the MKN-28 cells. Following incubation with *Hpcf*, the MKN-28 cells display increased concentration of BCAA, while the extracellular medium shows reduced concentration of BCAA (Figure 2A,B). Since both *Hp* and MKN-28 cells are auxotrophic for essential amino acids, the above interpretation seems plausible.

### 2.4. Helicobacter Pylori Induces Inflammation and Oxidative Stress

*Hp* infection induces inflammation via activation of *NF-kB*. In our study, cell inflammation is confirmed by the upregulation of the *IL-8*, *TNF-α*, *Il-6*, *TLR2*, and *TLR9* genes expressed by the MKN-28 cells incubated with *Hpcf* (Table 1). These cytokines characterize *Hp* infection and are part of a panel validated to detect inflammation in patients with chronic diseases [32]. If not controlled, inflammation may damage the gastric cells. In our study, we find several metabolites controlling the inflammation triggered by *Hp*. Taurine curbs the excess of Reactive oxygen species (ROS) produced by mitochondria [33]. The reduced level of this metabolite (Figure 3A) may indicate that part of it has been used to control the high levels of ROS produced by the mitochondria. One more signature in the same direction is sialic acid, expressed at a high level (Figure 3A). This metabolite assists the immune system to discriminate between the self and non-self [34]. Inflammation is also regulated by intracellular (acetate and propionate) and the extracellular (lactate) metabolites (Figure 3A,B). Acetate and propionate modulate the gastric mucosa inflammation [35]. Lactate, the product of the catabolism of glucose under anaerobic conditions, induces expression of anti-inflammatory genes, and promotes macrophage polarization [36].

High levels of BCAAs (Figure 2A) and FA (Figure 3B) cause accumulation of catabolic intermediates (propionyl CoA and succinyl CoA) and incompletely oxidized FA contribute to the mitochondrial stress [37]. PCR array technology displayed upregulation of the main genes involved in the cell oxidative stress (Table 1). In particular, upregulation of *SOD2*, and downregulation of *SOD1*, point out the involvement of mitochondria in the production of ROS (Figure 4). Mitochondrial dysfunction is further confirmed by the high level of glutamate, which is tightly correlated with mitochondrial stress [35] (Figure 3B).

BCAA are required for the activation of mTORC1 [38]. PCR array technology showed up-regulation of both the *RPTOR* and *MLST8* subunits of mTORC1 after 2 h of *Hpcf* treatment, with key genes implicated in the regulation of the mTORC1 (*AKT1*, *AKT2*, *INSR*, *IRS1*, *PLD1*, and *RPS6KA*2) being significantly up-regulated (*p*-value ≤ 0.0001). Significantly, upregulated genes include *AKT1*-involved in the inhibition of protein breakdown-and *INSR*-sensing the concentration of insulin outside the cell and transmits this signal through the *PI3K*/Akt/mTORC1 pathway [39]. Finally, mTORC1 activation is confirmed by the overexpression of the positive effectors *CHUK*, *EIF4E*, and *HIF1A* (Table 1).

We can conclude that high levels of BCAA (Figure 2A), activation of mTORC1, impaired mitochondrial activity, and inflammation (Table 1) are dominant traits common to the four diseases under investigation, and all potentially referable to *Hp*.

## 3. Discussion

Cellular homeostasis is heavily dependent upon a balanced regulation of mTORC1. Thus, deregulation of this complex inevitably leads to many diseases, including metabolic diseases. Though mTORC1 can be modulated by several factors, amino acids are condition necessary and sufficient for promoting cellular anabolic metabolism and mTORC1 activation [40]. Upon incubation with *Hpcf*, MKN-28 cells show high levels of amino acids (Figure 2A). This result, also observed in the AGS human cell line, reflects the demand of nutrients by *Hp* [41]. At the same time, it displays the tight connection linking mTORC1 activation, high amino acids concentrations, and *Hp* infection.

In the absence of inflammation, immune cells exploit an anabolic metabolism modulated by mTORC1 tempered by *c-MYC*. In the presence of inflammation, *c-MYC* is suppressed and mTORC1 moves under the control of *HIF1α* [42]. This finding indicates that cell replication and inflammation both depend upon mTORC1, but are regulated by *c-MYC* in the absence of inflammation and *H1F1α* in the presence of inflammation. In addition of inducing anabolic metabolism, inflammation inhibits *AMPK*, which is the activator of catabolic metabolism [43]. In our study, upregulation of *HIF1α* and mTORC1 indicates that MKN-28 cells, upon incubation with *Hpcf*, express the mTORC1-*HIF1α*-regulated inflammatory phenotype (Table 1). Induction of *TNF-α* and *IL-6* (Table 1) confirms the suppression of *AMPK*, which is a condition required for expression of these cytokines. Enhanced glycolysis via activation of *PIK3* subunits and mTORC1-*HIF1α* (Table 1) and suppression of *AMPK* further confirm the anabolic state of the MKN-28 cells. Activation of MKN-28 cells by *Hpcf* and expression of the genes modulating mTORC1-*HIF1α* are one more proof of the tight connection between *Hp* and mTORC1 activation.

As a supervisor of host cell nourishment, mTORC1 is also the target of bacteria. To establish infection, bacteria often need to shift the host cellular metabolism from anabolic to catabolic. Inhibition of mTORC1 by the *Hp* toxin VacA is interpreted as a means used by the bacterium to prevent the production of nutrients needed by immune cells of the host to mount an antibacterial response [12]. Taken alone, inhibition of mTORC1 by VacA may seem at odds with the activation of the same gene by increased concentrations of amino acids. Actually, *Hp* has adopted this strategy to better exploit the resources of the host. Autophagy induced by the VacA toxin is useful to *Hp* during the gastric mucosa colonization phase [44]. However, once *Hp* has colonized the gastric mucosa, autophagy becomes harmful for the survival of bacteria. Therefore, in this phase of infection, VacA is neutralized by CagA [10]. Two virulence factors of *Hp*, acting antagonistically, protect the survival of bacteria during different phases of the infection: a highly rewarding result for *Hp* that, by the same mechanism, can inhibit or activate the host cellular metabolism.

*Hp* directly impacts on mitochondria and inflammation. Mitochondria are involved in ATP production, apoptosis, lipid, and amino acids metabolism [45]. In our in vitro experiment, the signatures of mitochondrial stress are evident as upregulation of the antioxidant superoxide dismutase *SOD2* (Table 1) and those of inflammation as secretion of the pro-inflammatory factors as well as *TNF-α* and *IL-6* genes (Table 1). At this stage, we can conclude that the traits common to the four diseases can all be associated, at least in part, with *Hp*.

Several interesting insights emerge from this study. First, it demonstrates that cell lines can be used successfully to investigate genetic or metabolic disorders. Cell line experiments, compared to studies on patients, offer the advantage of reducing variables, differences in the diet, use of drugs, or genetic heterogeneity, between participants. Second, the study raises the question whether a concerted effort to study the basic biology of diseases with several traits in common (as T2D, OB, AD, and CMT) may be more effective than the traditional approach of studying each disease separately. Third, here, we point out that *Hp* and mammals have lost the genes coding for essential amino acids. Evolution gives some hints about the benefits associated with gene loss [46]. Organisms, bacteria as well as mammals, evolve under changing environmental conditions. If amino acids are available in the environment, the corresponding genes, no longer adaptive, are lost or undergo mutations leading to more adaptive genes. Following this logic, it seems clear that mammals and *Hp* are auxotrophic for essential amino acids. Mammals can obtain essential amino acids from their diet. *Hp* finds them in its niche (the gastric mucosa), which is rich in essential amino acids derived from the diet of the host [7].

Finally, we acknowledge a limit of this study. We describe four traits common to the four diseases and *Hp* infection. However, this finding is not sufficient to attribute to *Hp* a causal role for the above diseases. *Hp* could just be the biomarker of a genetic or metabolic disorder carried by the MKN-28 cell line. It is hard to distinguish between causation and association, especially in the case of highly complex diseases. Despite the fact that high levels of BCAA anticipate of many years T2D, it is not known yet whether BCAA are the cause of T2D or a biomarker of insulin resistance [11]. However, human stem cells-derived organoids [47] from healthy donors and from patients with the above diseases may permit us to repeat the study in a known genetic context and clarify the role of *Hp*.

## 4. Materials and Methods

### 4.1. Helicobacter Pylori

*Hp* strain ATCC 43504 was grown in Brain Heart Infusion medium (BHI, Oxoid, UK) complemented with 10% Fetal Bovine Serum (FBS, Oxoid, UK) and incubated under microaerophilic condition at 37 °C [48].

### 4.2. Cell Culture Conditions

The human gastric adenocarcinoma MKN-28 cell lin00000000000e (ATCC, MD, USA) was grown in DMEM/F12 medium, supplemented with 10% FBS, 1% penicillin/streptomycin, and 1% glutamine in a 100-mm culture dish at 37 °C in a 5% CO_2_ atmosphere. The reagents were all from Gibco, ThermoFisher, Waltham, MA, USA.

### 4.3. Cell Culture for Metabolomics Analysis

Confluent 100-mm dishes were incubated for two hours with *Hpcf*, or as a control with BHI medium. The cells were detached from the adherent substrate with trypsin (1.5 mL for 3 min), and washed twice with Phosphate-Buffered Saline (PBS). The mixture, containing detached cells, was transferred into a Falcon tube and centrifuged at 1200 rpm and 25 °C for 3 min. The liquid phase was discarded, and the cell pellet was washed three times with 5 mL of PBS. After the last wash, the cell pellet was frozen in liquid nitrogen and stored at −80 °C until metabolites extraction.

### 4.4. Metabolites Extraction and NMR Samples Preparation

Combined extraction of polar and hydrophobic metabolites was carried out by using ice-cold methanol/Milli-Q^®^ water/chloroform (1/0.72/1) solvents as described by Papaianni et al. 2020 [49]. NMR-based intracellular metabolomics analysis and NMR-based extracellular metabolomics analysis were shown as Papaianni et al., 2020 [50].

### 4.5. NMR Spectroscopy

NMR spectra were recorded on a Bruker Avance III-600 MHz spectrometer (Bruker BioSpin GmbH, Rheinstetten, Germany) as reported [51].

### 4.6. RNA Extraction and Real-Time PCR

Gene expression analysis through PCR array technology was performed as described [52]. Briefly, cells (2 × 10^6^), used for RNA extraction, were seeded in Petri dishes (100 mm diameter) and treated with *Hpcf*. After 2 h of exposure time, cells were washed directly in the Petri dish by adding cold PBS. Cells were lysed in the Petri dish by adding 1 mL of Trisure Reagent (Bioline, Memphis, TN, USA). RNA was isolated according to the manufacturer’s protocol. RNA concentration and purity were assessed using the nanophotomer NanodroP (Euroclone, Milan, Italy). RNA (200 ng) was reverse transcribed using the RT2 first strand kit (Qiagen, Hilden, Germany), according to the manufacturer’s instructions. The qRT-PCR analysis was performed in triplicate using the RT2 Profiler PCR Array kit (Qiagen, Hilden, Germany). Plates were run on a ViiA7 (Applied Biosystems, Foster City, CA, USA) according to the Standard Fast PCR Cycling protocol with 10 μL reaction volumes. Cycling conditions were: 1 cycle initiation at 95.0 °C for 10 min, which was followed by amplification for 40 cycles at 95.0 °C for 15 s and 60.0 °C for 1 min. Amplification data were collected via ViiA 7 RUO Software (Applied Biosystems, Foster City, CA, USA). The cycle threshold (Ct)-values were analysed with PCR array data analysis online software (http://pcrdataanalysis.sabiosciences.com/pcr/arrayanalysis.php, Qiagen, Hilden, Germany).

## Figures and Tables

**Figure 1 ijms-21-08369-f001:**
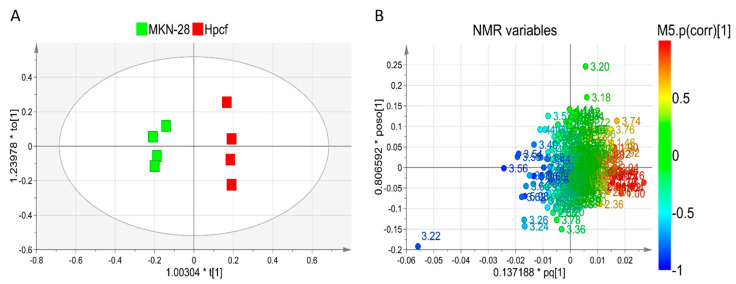
OPLS-DA of MKN-28 cell samples. (**A**) Scores plot showing the distinct separation between MKN-28 cells incubated (red squares) and not incubated (green squares) with *Hpcf*. The X-axis title represents the factor multiplied (*) the principal predictive component t[1] which better approximates the dataset variation correlated with samples classification. On the ordinate axis, the title represents the factor multiplied (*) the orthogonal component t[1] which accounts for intraclass variation. (**B**) Loadings plot of nuclear magnetic resonance (NMR) variables (chemical shift) referred to metabolites responsible for between-classes separation and characterized by |*p*(corr)| value > 0.7.

**Figure 2 ijms-21-08369-f002:**
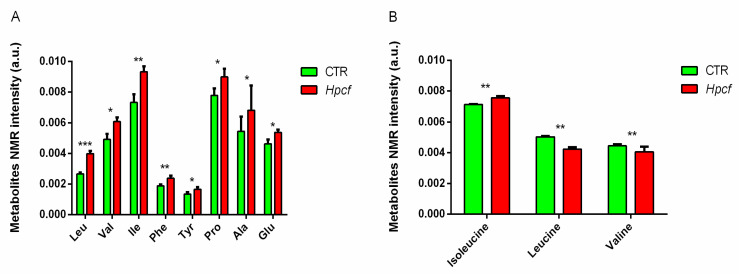
(**A**) Intracellular amino acid concentration differences (leucine, valine, isoleucine, phenylalanine, tyrosine, proline, and alanine) detected in MKN-28 cells incubated (red columns) or not incubated (green columns) with *Hpcf*. (**B**) Extracellular BCAA (leucine, isoleucine, and valine) concentration differences detected in culture medium of MKN-28 cells incubated (red columns) or not incubated (green columns) with *Hpcf*. Some *Hp* strains synthesizes isoleucine. Our strain is one of the strains that justify its upregulation. The X-axis reports a single amino acid and the Y-axis reports the bucket variation corresponding to the specific amino acid scaled to the total NMR spectral area. Intensity of amino acids is expressed in arbitrary units and represented as means ± SD (* *p* < 0.5, ** *p* < 0.01, *** *p* < 0.001) calculated from two experiments in which each is performed in quadruplicate.

**Figure 3 ijms-21-08369-f003:**
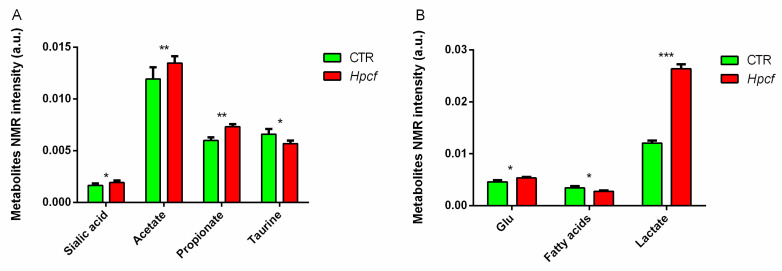
(**A**) Metabolite concentration differences (sialic acid, acetate, propionate, and taurine) detected in MKN-28 cells incubated (red columns) or not incubated with *H. pylori* cell filtrate (*Hpcf*). (**B**) Metabolite concentration differences (glutamate, fatty acids, and lactate) detected in MKN-28 cells incubated (red columns) or not incubated with *H. pylori* cell filtrate (*Hpcf*). The X-axis reports a single metabolite and the Y-axis reports the bucket variation corresponding to the specific metabolite scaled to the total NMR spectral area. Intensity of metabolites is expressed as arbitrary units and represented as means ± SD (* *p* < 0.5, ** *p* < 0.01, *** *p* < 0.001) calculated from two experiments with each performed in quadruplicate.

**Figure 4 ijms-21-08369-f004:**
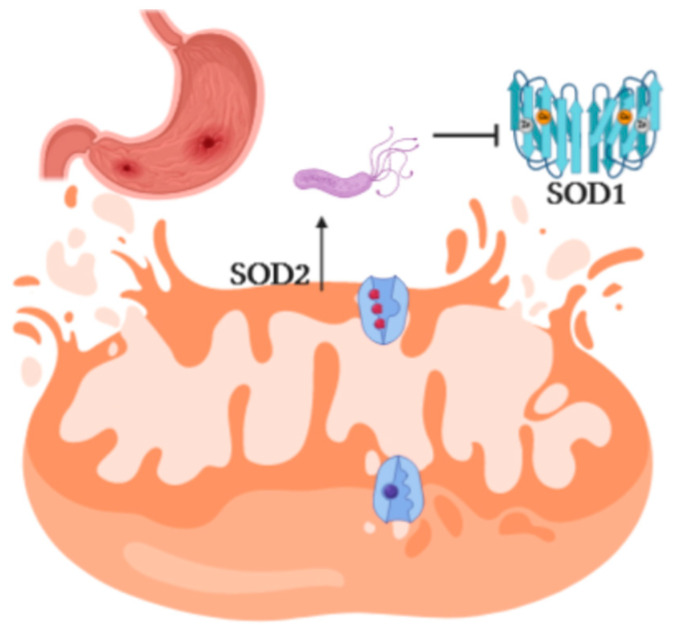
Schematic representation of superoxide dismutase 1 (*SOD1*) and superoxide dismutase 2 (*SOD2*) genes modulated by *Hp*. Downregulation of *SOD1* and upregulation of *SOD2* point out the involvement of mitochondria in ROS production.

**Table 1 ijms-21-08369-t001:** Genes of mammalian Target Of Rapamycin (*mTOR*) signaling, inflammatory, and oxidative stress pathways detected by polymerase chain reaction (PCR) array technology and differently expressed in MKN-28 cells incubated with *Hpcf* for 1 or 2 h. Variation of gene expression levels is reported as fold regulation. Values > |2| are considered statistically significant.

Pathway Name	Gene ID	Gene Name	Fold Regulation 1 h	Fold Regulation 2 h
***mTOR signaling pathway***	***RPTOR***	Regulatory associated protein of mTOR complex 1	−1.42	286.04
	***MLST8***	mTOR associated protein, LST8 homolog (S. cerevisiae)	−1.42	398.95
	***AKT1***	V-akt murine thymoma viral oncogene homolog 1	−1.42	50.13
	***AKT2***	V-akt murine thymoma viral oncogene homolog 2	−1.42	504.97
	***INSR***	Insulin receptor	−1.42	257.79
	***IRS1***	Insulin receptor substrate 1	−1.42	278.22
	***PLD1***	Phospholipase D1, phosphatidylcholine-specific	−6.31	130.70
	***RPS6KA2***	Ribosomal protein S6 kinase, 90 kDa, polypeptide 2	−1.24	3.37
	***PDPK1***	3-phosphoinositide dependent protein kinase-1	−1.53	28.25
	***PIK3CB***	Phosphoinositide-3-kinase, catalytic, beta polypeptide	−1.42	16.34
	***PIK3CD***	Phosphoinositide-3-kinase, catalytic, delta polypeptide	3.37	184.83
	***PIK3CG***	Phosphoinositide-3-kinase, catalytic, gamma polypeptide	−1.42	215.28
	***CHUK***	Conserved helix-loop-helix ubiquitous kinase	−4.08	181.03
	***EIF4E***	Eukaryotic translation initiation factor 4E	−1.42	922.92
	***HIF1A***	Hypoxia inducible factor 1, alpha subunit	192.93	955.47
***Inflammatory pathway***	***CXCL8***	Interleukin 8	−3.29	2.96
	***IL-6***	Interleukin 6	14.45	114.56
	***TLR2***	Toll-like receptor 2	58	72.18
	***TLR9***	Toll-like receptor 9	3.29	134.55
	***TNF***	Tumor necrosis factor	12.9	154.26
***Oxidative stress pathway***	***ATOX1***	ATX1 antioxidant protein 1 homolog (yeast)	3.57	37.69
	***GPX2***	Glutathione peroxidase 2 (gastrointestinal)	3.57	37.69
	***GPX4***	Glutathione peroxidase 4 (gastrointestinal)	3.57	37.69
	***GSS***	Glutathione synthetase	3.57	9.54
	***NOX5***	NADPH oxidase. EF-hand calcium binding domain 5	3.57	7.54
	***SOD1***	Superoxide dismutase 1. soluble	−28.68	−9.67
	***SOD2***	Superoxide dismutase 2. mitochondrial	3.96	4.04

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
