# Peer review of "An In Vitro Model to Investigate the Role of Helicobacter pylori in Type 2 Diabetes, Obesity, Alzheimer’s Disease and Cardiometabolic Disease"

_ijms, 2020, doi:10.3390/ijms21218369_

Round 1
Reviewer 1 Report
- There is no explanation for the limitation in discussion part.
- Authors should present some evidence to support their arguments. Please mention about causal relation between four diseases (T2D, OB, AD, CMD) and HP.
- There is a need for further explanations of what the purpose of revealing that HP modulate these traits; upregulation of mTORC1, inflammation, and mitochondrial dysfunction.
- The order of methods and results in Article has been changed. Please correct it.
Author Response
- There is no explanation for the limitation in discussion part.
Please see last paragraph of the discussion. .
- Authors should present some evidence to support their arguments. Please mention about causal relation between four diseases (T2D, OB, AD, CMD) and HP.
In the present study we describe four traits (mTORC1 upregulation, high levels of BCAA, inflammation, and mitochondrial dysfunction) common to the above diseases and Hp infection. Still at present we cannot attribute to Hp a causal role for the above diseases. Hp could just be the biomarker of a genetic defect carried by the MKN-28 cell line or any unknown confounding factor. It is hard to distinguish between causation and association, especially in the case of highly complex diseases. Despite the fact that high levels of BCAA anticipate of many years T2D, it is not known yet whether BCAA are the cause of T2D or a biomarker of insulin resistance (Nat Rev Endocrinol 10; 723). Human stem cells-derived organoids (Gastroenterology 148; 126) from healthy donors and from patients with the above diseases may permit us to repeat the study in a known genetic context and thus clarify the role of Hp.
- There is a need for further explanations of what the purpose of revealing that HP modulate these traits; upregulation of mTORC1, inflammation, and mitochondrial dysfunction.
High levels of BCAA, mTORC1 upregulation, inflammation, and mitochondrial dysfunction – as traits common to T2D, OB, AD, CMD and Hp – were plausibly expected to provide evidence that Hp is a causal factor of the above diseases. Our expectation did not occur. However, the present study has suggested us next experiment to do.
- The order of methods and results in Article has been changed. Please correct it.
Methods were placed after Discussion, as indicated in the section “Instruction for the authors” of the journal.

Reviewer 2 Report
The manuscript describes some in vitro experiments aimed to investigate the role of Helicobacter pylori in chronic diseases like type 2 diabetes, obesity, AD, and cardiometabolic disease. Upregulation of mTORC1, inflammation, and mitochondrial dysfunction were found showing how Hp may manipulate the main traits common to the above-mentioned diseases. This is a carefully designed study forming an important step in our understanding of the pathogenesis of the civilization diseases.
Major comments
References should be avoided in Results section. In its current form, the Results section is too reminiscent of a Discussion.
Aim of the study should be clearly stated at the end of Introduction.
Conclusions should be clearly stated at the end of Discussion.
Minor comments
Abstract – abbreviations must be defined in first use (mTORC1).
Figure 1 – abbreviations must be defined in figure legend (MKN, Hpcf). Titles of the y axis need to be corrected.
Figure 2 – abbreviations must be defined in figure legend (CTR, Hpcf, BCAA).
Figure 3 – abbreviations must be defined in figure legend (CTR, Hpcf, MKN).
Figure 4 – abbreviations must be defined in figure legend (SOD1, SOD2).
Table 1 - abbreviations must be defined (mTOR and others).
Author Response
Response to Reviewer 2 Comments
References should be avoided in Result section. In the current form, the results section is too reminiscent of discussion.
The references of the Results section have been strictly limited to those providing relevant data that the reader may be interested to study. The first part of the Discussion - overlapping with the Results – has been omitted.
Aim of the study should be clearly stated at the end of Introduction.
Please see the third paragraph of the Introduction. .
Conclusions should be clearly stated at the end of Discussion.
Please see the last paragraph of the Discussion. .
Minor comments
Abstract – abbreviations must be defined in first use (mTORC1).
Figure 1 – abbreviations must be defined in figure legend (MKN, Hpcf). Titles of the y axis need to be corrected.
Figure 2 – abbreviations must be defined in figure legend (CTR, Hpcf, BCAA).
Figure 3 – abbreviations must be defined in figure legend (CTR, Hpcf, MKN).
Figure 4 – abbreviations must be defined in figure legend (SOD1, SOD2).
Table 1 - abbreviations must be defined (mTOR and others).
Abbreviations have all been defined in the proper legends or when used for the first time, as suggested. MKN-28 is not an abbreviation, but the acronym of the cell line. In addition, titles of the y axis of Figure 1 has been corrected.

Round 2
Reviewer 1 Report
This paper may be published.